# Antistars or Antimatter Cores in Mirror Neutron Stars?

**Zurab Berezhiani** [1,2]

1 Dipartimento di Fisica e Chimica, Università di L'Aquila, 67100 Coppito, L'Aquila, Italy; zurab.berezhiani@aquila.infn.it
2 INFN, Laboratori Nazionali del Gran Sasso, Assergi, 67010 L'Aquila, Italy

**Abstract:** The oscillation of the neutron $n$ into mirror neutron $n'$, its partner from the dark mirror sector, can gradually transform an ordinary neutron star into a mixed star consisting in part of mirror dark matter. The implications of the reverse process taking place in the mirror neutron stars depend on the sign of baryon asymmetry in the mirror sector. Namely, if it is negative, as predicted by certain baryogenesis scenarios, then $\bar{n}' - \bar{n}$ transitions create a core of our antimatter gravitationally trapped in the mirror star interior. The annihilation of accreted gas on such antimatter cores could explain the origin of $\gamma$-source candidates with an unusual spectrum compatible with baryon–antibaryon annihilation, recently identified in the Fermi LAT catalog. In addition, some part of this antimatter escaping after the mergers of mirror neutron stars can produce the flux of cosmic antihelium and also heavier antinuclei which are hunted in the AMS-02 experiment.

**Keywords:** dark matter; mirror matter; neutron oscillation; antimatter

## 1. Introduction

Ordinary baryons account only for about 20% of matter in the Universe while the remaining 80% is represented by a hypothetical form of matter known as dark matter (DM), the presence of which is suggested by a variety of astrophysical and cosmological observations. The DM identity remains unknown: the Standard Model (SM) in itself suggests no candidate, the search for DM particles at colliders remains unsuccessful up to now and there are no confirmed signals for their direct or indirect detection.

One of the possible DM candidates is mirror (M) matter, represented by the particles of parallel hidden sector which are exact replicas of ordinary (O) particles: mirror parity, a discrete symmetry under the specular exchange of O and M species (fermion, gauge and Higgs fields of two sectors), implies that the two sectors should have exactly identical microphysics (for reviews see e.g., [1–3]). Hence, all O particles: electron $e$, proton $p$, neutron $n$, neutrinos $\nu$, photon $\gamma$ etc. have their mass-degenerate M twins $e'$, $p'$, $n'$, $\nu'$, $\gamma'$ etc. which are sterile to the SM interactions, but have their own SM$'$ interactions with exactly the same pattern. Moreover, since the O matter fraction $\Omega_B$ in the Universe is related to the baryon asymmetry (BA) produced by some baryogenesis mechanism *a lá* Sakharov [4], the mirror matter fraction $\Omega'_B$ should be produced by analogous baryogenesis in M sector. Thus, M matter represents a DM of asymmetric type and, alike the O matter, during the cosmological evolution it should form the mirror nuclei, atoms, molecular clouds, stars etc. which are dark in terms of O photons.

A specific feature of this scenario is that any neutral particle, elementary (as e.g., neutrinos) or composite (as e.g., the neutron) can have a mixing with its mass degenerate M partner. Namely, the active–sterile oscillations $\nu - \nu'$ between the O and M neutrinos, violating the lepton numbers L and L$'$ of both sectors by one unit, can be experimentally observed as the deficit of neutrinos.

Analogously, the mass mixing between the neutron $n$ and its dark partner $n'$, mirror neutron [5,6],

$$\varepsilon \, \overline{n'} n + \text{h.c.,} \tag{1}$$

violates the baryon numbers B and B′ by one unit but conserves the combination $\widetilde{B} = B + B′$. This mixing induces the neutron–mirror neutron ($n − n′$) oscillation which is similar to that of the neutron–antineutron ($n − \bar{n}$) oscillation violating B by two units [7,8], and these two phenomena can have a common origin in the context of certain theoretical models [5,9]. However, there is a drastic difference. Namely, for $n − \bar{n}$ mixing the direct experimental limit on $n − \bar{n}$ oscillation time $\tau_{n\bar{n}} = 0.86 \times 10^8$ s implies the upper bound $\varepsilon_{n\bar{n}} < 7.7 \times 10^{-24}$ eV while the bound from the nuclear stability is even stronger, $\varepsilon_{n\bar{n}} < 2.5 \times 10^{-24}$ eV or so (for a review of see [10–12]).

On the other hand, as it was shown in [5,6], non of the existing phenomenological or cosmological bounds can exclude $n − n′$ mixing mass in (1) to be as large as $\varepsilon \sim 10^{-15}$ eV which corresponds to the oscillation time $\tau_{nn′} = 1/\varepsilon$ of few seconds (in this paper we use natural units $c = 1$ and $\hbar = 1$). For free neutrons $n − n′$ oscillation is suppressed by the medium effects as the presence of matter and magnetic fields [5,6]. For $\tau_{nn′}$ smaller than the neutron decay time, $n − n′$ oscillation can have interesting astrophysical implications for ultra-high energy cosmic rays [5,13,14] and for the neutrons from solar flares [15], and it can be experimentally searched at the existing neutron facilities via the neutron disappearance $n \to n′$ and regeneration $n \to n′ \to n$ [5,6,16,17]. In fact, the ultra-cold neutron (UCN) losses induced by $n − n′$ oscillations were already measured in several dedicated experiments [18–24], and they still do not exclude rather short oscillation times. Moreover, some of these experiments show deviations from null hypothesis [18,23,25], and new experiments are underway for testing these anomalies [26–28]. The cosmological limits on $n − n′$ mixing were discussed in [5] and recently in [29].

Remarkably, the nuclear stability implies no bounds on $\varepsilon$ since $n \to n′$ conversion for the neutrons bound in nuclei by nuclear forces is forbidden by the energy conservation [5]. However, in the neutron stars (NS) bound by gravity $n \to n′$ conversion is energetically favored, and it can gradually transform the NS into the mixed stars partially consisting of M matter [5]. The effects of such a transformation on the NS masses and radii, on their surface temperatures and on the pulsar timing were discussed in Refs. [30–32] and several related effects were recently considered in Refs. [33–38].

By the essence of the mirror matter, the neutron stars should exist also in dark M sector. Therefore, the mirror NS should reciprocally develop the similar fraction of O matter in their interiors. However, there is a caveat: the sign of mirror BA is a priori unknown. (Notice that by naming $n′$ as mirror neutron, we implicitly extend the notion of our baryon charge B to that of M species B′ classifying the latter via the combined charge $\widetilde{B} = B + B′$ which is conserved by $nn′$ mixing (1). In fact, the M species with $\widetilde{B} = 1$ and $\widetilde{B} = −1$ for us respectively are the mirror baryons (MB) and anti-mirror baryons (AMB). In particular, $\bar{n}′$ is anti-mirror neutron or mirror antineutron, no matter how they can be qualified by the inhabitants of M world).

The sign of ordinary BA, $\mathcal{B} = \text{sign}(n_b − n_{\bar{b}}) = 1$, is fixed by some baryogenesis mechanism [4] which created the primordial excess of baryons over antibaryons. At first glance, the sign of mirror BA should be the same by M parity, $\mathcal{B}′ = \text{sign}(n_{b′} − n_{\bar{b}′}) = 1$. This would be the case if the identical baryogenesis mechanisms act separately in the O and M sectors. However, one can also envisage a co-baryogenesis scenario e.g., via the processes which violate B and B′ but conserve $\widetilde{B} = B + B′$ (which processes can in turn be related to some new physics that induces $nn′$ mixing (1) itself). In this case, for $\mathcal{B}$ being positive, the null O+M asymmetry in the Universe would imply the negative $\mathcal{B}′$. The co-genesis models which induce BAs in both sectors via the B−L and B′−L′ violating cross-interactions between the O and M particle species indeed predict $\mathcal{B}$ and $\mathcal{B}′$ of the opposite signs [39,40].

Hence, depending on the sign of $\mathcal{B}′$, the compact stars in M sector can be the mirror neutron stars (MNS) or anti-mirror neutron stars (AMNS) consisting respectively of the MB or AMB. Correspondingly, $n′ \to n$ transitions in the MNS (or $\bar{n}′ \to \bar{n}$ in the AMNS) can create the cores of ordinary matter (or antimatter) in their interior. In both cases these cores can be detectable as hot compact sources emitting the photons in the far UV and X-ray

ranges [31] but the electromagnetic emission cannot trace the composition of their cores (matter or antimatter). However, the presence of interstellar medium makes the difference. In the case of the MNS accretion of interstellar gas can only heat its O matter core and induce X-ray emission, whereas in the case of AMNS the accreted gas will annihilate on the surface of its antimatter core producing gamma-rays with energies up to a GeV or so.

Interestingly, the recent analysis [41] based on the 10-year Fermi Large Area Telescope (LAT) gamma-ray source catalog [42] has identified 14 point-like candidates emitting $\gamma$s with a spectrum compatible to baryon–antibaryon annihilation [43], and not associated with objects belonging to established gamma-ray source classes.

In this letter we discuss the possibility whether the unusual sources of this type can be the AMNS with the antimatter cores, which produce the annihilation $\gamma$-rays by accretion of interstellar gas, and whether some part of this antimatter can escape from the AMNS producing the antinuclei detected by AMS-02 in cosmic rays.

The paper is organised as follows. In the next section we discuss the concept of mirror sector and mirror matter as dark matter. Section 3 discusses the $B - L$ violating phenomena between the O and M particles and the baryogenesis mechanisms in two sectors, namely their implications for the sign of mirror BA. In Section 4 we discuss $n - n'$ transitions in neutron stars and evaluate the transition time. Section 5 discusses $\bar{n}' - \bar{n}$ transitions in mirror (anti)neutron stars which can create ordinary antimatter cores in their interior, and its implications for baryon–antibaryon annihilation signals and mirror neutron star mergers. Finally, in Section 6 we summarize our findings in a more general context and discuss how the signals of antimatter cores in mirror neutron stars can be distinguished from those of antistars.

## 2. Mirror Matter as Dark Matter

Let us discuss our scenario in wider context, and take a pleasant walk and pleasant talk with the Walrus and the Carpenter for viewing and reviewing a panorama of two parallel worlds. This picture is based on the direct product $G \times G'$ of two identical gauge groups represented by the SM or some its extension, with O particles belonging to $G$ and M particles to $G'$. The total Lagrangian is

$$\mathcal{L}_{\text{tot}} = \mathcal{L} + \mathcal{L}' + \mathcal{L}_{\text{mix}}, \tag{2}$$

where $\mathcal{L}$ and $\mathcal{L}'$ stand respectively for the O and M sectors, whereas the mixed Lagrangian $\mathcal{L}_{\text{mix}}$ describes the possible cross-interactions between the particles of two sectors. The identical forms of $\mathcal{L}$ and $\mathcal{L}'$ is ensured by a discrete symmetry $G \leftrightarrow G'$ interchanging all O species (the fermion, gauge and Higgs fields of $G$ sector) with their M partners (the fermion, gauge and Higgs fields of $G'$ sector). In the context of extra dimensions, it can be viewed as a geometric symmetry between two parallel 3-branes hosting the O and M particle species. In principle, one can also consider the models with more than one parallel sectors.

M baryons as cosmological DM have specific cosmological implications [44–53]. Although O and M components have identical microphysics, their cosmological realizations must be quite different. Namely, the viability of M sector requires the following conditions [47,50]:

– *Asymmetric reheating*: after inflation the O and M sectors are reheated asymmetrically, with $T > T'$, which can be realized in certain inflation models [47,50,54];
– *Out-of-equilibrium*: interactions between O and M particles are feeble enough in order to maintain the temperature asymmetry in subsequent epochs. In other words, all cross-interactions in $\mathcal{L}_{\text{mix}}$ should remain out-of-equilibrium at any stage after inflation, at least before the Big Bang Nucleosynthesis (BBN);
– *No extra heating*: after inflation both sectors evolve almost adiabatically and the temperature difference $T' < T$ is not erased by the entropy production in M sector e.g., due to the first order phase transitions.

Namely, the BBN bounds require $T' < T/2$ [50] while the post-recombination cosmology is yet more restrictive, demanding $T' < T/4$ or so [52]. This has interesting consequences for the primordial chemical content in M world: while O world is dominated by hydrogen, with it primordial mass fraction being 75% and that of helium being 25%, M world should instead be helium dominated, with the mass fractions of M hydrogen and M helium being, respectively, 25% and 75% or so [50].

Along with the ordinary stars, dark mirror stars can also be formed in the Galaxy. However, since M world is colder, the first mirror stars must start forming somewhat earlier than the first (population III) stars in the O sector. The M stars, being helium dominated, should have somewhat different initial mass function and their evolution should be more rapid as compared to O stars [53]. Since helium is less opaque than hydrogen, the massive M stars should suffer less mass losses from stellar winds than their O analogues, and most of them can end up collapsing into massive black holes (BH). The intermediate mass M stars can explode as SN and form mirror neutrons stars, while M stars of the solar mass can survive up to present times. All these objects, together with a some fraction of mirror gas, can constitute dark matter in the Galaxy. (Since mirror gas is a self-interacting type of DM, its fraction should be subdominant by astrophysical restrictions from Bullet cluster etc.) Namely, the galactic halo can be viewed as elliptical mirror galaxy consisting of M stars (and the BH originated from the collapse of heavy M stars) in which O matter forms the disc [50]. M matter could also contribute to the disc, but the density of M stars in the disc cannot exceed the density of O stars [55]. All these objects can be observed via microlensing as the Machos in different mass ranges. The present limits from EROS-MACHO observations do not exclude the possibility of the galactic halo dominated by dark objects as the BH with masses $M > (10 \div 100) \, M_\odot$ while the fraction of dark stars with $M \sim M_\odot$ can be $\sim 10\%$ or so. This proportion can correspond to the abundance of the LIGO gravitational wave (GW) signals [56] from the BH mergers with typical masses $M \sim (10 \div 50) \, M_\odot$ or so [57,58]. In addition, some of the peculiar LIGO events with one or both light components and no optical counterpart can be viewed as the MNS-MNS or BH-MNS mergers [59–61].

On the other hand, there are no fundamental reasons to think that two sectors should be connected only by a common gravity. (Moreover, the picture can be extended to bigravity scenarios, where at short distances O and M components have different gravities [62,63] with one combination of gravitons getting the Lorentz-breaking masses terms [64].) The possible cross-interactions $\mathcal{L}_{\mathrm{mix}}$ between the O and M particles can provide specific portals for the detection of DM in the form of mirror gas, and can induce the mixing phenomena between the O and M species. For example, kinetic mixing between the O and M photons, $\varepsilon F^{\mu\nu} F'_{\mu\nu}$ [65], induces the positronium oscillation into M positronium [66] and, on the other hand, can provide a portal for the identification of mirror atoms in DM detectors [67,68] and can also explain the origin of galactic magnetic fileds [69]. The possibility of dark photon kinetic mixing was also discussed in the literature [70]. The particles of two sectors may interact also through the gauge bosons of e.g., the common family symmetry $SU(3)_H$ [71,72] or common $U(1)_{\mathrm{B-L}}$ symmetry [73], and these interactions can lead to mixing of neutral O and M mesons. The two sectors can also share the Peccei–Quinn symmetry, with the axion interacting with both O and M species [74]. An interesting link between two sectors can be provided also by micro black holes [75].

However, the most interesting interactions in $\mathcal{L}_{\mathrm{mix}}$ are the ones that violate baryon and lepton numbers of both sectors, such as, e.g., the ones which from one hand can induce the mixings between the neutrinos and/or the neutrons of two sectors, and on the other hand can induce baryon asymmetries in both O and M worlds.

## 3. Sign of Mirror Baryon Asymmetry

Now the time has come to talk of many things: of shoes and ships and sealing-wax, of cabbages and kings...; there is a subtlety related to the chiral character of the fermion representations in the SM: in our weak interactions the fermions are left-handed (LH) whereas the antifermions are right-handed (RH), and in the absence of CP-violating effects

two systems would be symmetric. In fact, we coin the LH-interacting species as fermions because we do consist of them since the sign of BA in the Universe, $\mathcal{B} = \mathrm{sign}(n_b - n_{\bar{b}}) = 1$, was fixed by a baryogenesis mechanism which created primordial excess of the baryons over antibaryons due to CP-violation in out-of-equilibrium processes violating B [4] and/or B − L [76]. In parallel sector the situation is same, apart of an ambiguity in the CP-violation pattern distinguishing between the M fermions and M antifermions, and the sign of mirror BA $\mathcal{B}' = \mathrm{sign}(n_b - n_{\bar{b}'})$. This ambiguity is related to the fact that $G \leftrightarrow G'$ symmetry can be realized in two ways: *with* or *without* chirality change between the O and M species [1,2].

Namely, three families of O fermions $f_{L,R}$ are described by the Weyl spinors in certain representations of gauge symmetry $SU(3) \times SU(2) \times U(1)$ of the SM, the LH quarks $q_L = (u,d)_L$ and leptons $\ell_L = (\nu,e)_L$ being weak doublets and the RH quarks $u_R, d_R$ and leptons $e_R$ being weak singlets (the family indices are suppressed). We assign the *positive* baryon and lepton numbers to these fields: B = 1/3 to $q_L, u_R, d_R$, and L = 1 to $\ell_L, e_R$. The antifermion fields obtained by complex-conjugation, $\bar{f}_{R,L} = C\gamma_0 f_{L,R}^*$, have opposite chiralities and opposite quantum numbers: $\bar{q}_R = (\bar{u}, \bar{d})_R$ and $\bar{u}_L, \bar{d}_L$ are antiquarks (B = −1/3), and $\bar{\ell}_R = (\bar{\nu}, \bar{e})_R$ and $\bar{e}_L$ are antileptons (L = −1). The symmetry between the fermions and antifermions, i.e., the invariance under CP transformation $f_{L,R} \to \bar{f}_{R,L}$, is violated by their complex Yukawa couplings to the SM Higgs doublet $\phi$.

As for the three mirror families represented by the Weyl spinors $f'_{R,L}$ in analogous representations of $SU(3)' \times SU(2)' \times U(1)'$ (SM′), we can invert the denominations coining as M fermions the species with the RH weak interactions, and assign them *positive* quantum numbers. Namely, to mirror quarks $q'_R = (u',d')_R$, $u'_L, d'_L$ we assign B′ = 1/3 and to mirror leptons $\ell'_R = (\nu',e')_R$, $e'_L$ we assign L′ = 1. Then the respective anti-fields, M antiquarks $\bar{q}'_L = (\bar{u}', \bar{d}')_L$, $\bar{u}'_R, \bar{d}'_R$ (B′ = −1/3) and M antileptons $\bar{\ell}'_L = (\bar{\nu}', \bar{e}')_L$, $\bar{e}'_R$ (L′ = −1) must have the LH weak interactions. (Once again, this is just a convention, and we could re-name these species in the opposite way). Needless to say, M fermions and M antifermions are equivalent modulo CP violating phases in their Yukawa couplings to the SM′ Higgs doublet $\phi'$.

Now, one can impose a discrete symmetry $\mathcal{Z}_2$ between the twin species of the *same chirality*: $f_{L,R} \leftrightarrow \bar{f}'_{L,R}$ which exchanges each O fermion with its M antifermion counterpart. Alternatively, we can employ $\mathcal{Z}_2^{LR} = CP\mathcal{Z}_2$ under the exchange $f_{L,R} \leftrightarrow f'_{R,L}$ between the O and M fermions having the *opposite chiralities*. (Clearly, both of these transformations should be complemented also by a proper exchange of the gauge and Higgs fields between two sectors.) Both of $\mathcal{Z}_2$ and $\mathcal{Z}_2^{LR}$ ensure the identical form of Lagrangians $\mathcal{L}$ and $\mathcal{L}'$ in (2), modulo the CP-violating phases in the Yukawa constants of two sectors.

In the case of $\mathcal{Z}_2^{LR}$ the 'right-handed' M matter should have exactly the same CP-violation pattern as our 'left-handed' matter, whereas for $\mathcal{Z}_2$ this equivalence holds for the 'left-handed' M antimatter. In the former case P parity, a symmetry between the 'left' and 'right', which is maximally violated in weak interactions of each sector, is in some sense restored between two sectors. In fact, this was the original motivation for introducing mirror fermions [77–79] (for a historical overview see also Ref. [80]). However, the real difference is related to CP-violation which was discovered just slightly before the original work [78]. In the absence of CP-violating phases in the Yukawa couplings $\mathcal{Z}_2$ and $\mathcal{Z}_2^{LR}$ symmetries would be equivalent. More generally, there could exist more than one parallel sectors corresponding to a direct product $G \times G_1 \times G_2 \times ...G_n$ of identical gauge factors. For any number $n$ of parallel sectors $\mathcal{Z}_2$ symmetry can be extended to a permutation symmetry $\mathcal{S}_n$ which will ensure that all sectors have identical CP pattern in the LH basis. For the even number $n = 2k$ instead one could impose a symmetry $\mathcal{S}_k \times \mathcal{S}'_k \times \mathcal{Z}^{LR}$ on the theory $[G \times G_1 \times ...G_k] \times [G' \times G'_1 \times ...G'_k]$ between $k$ LH and $k$ RH sectors with the identical CP pattern.

Now regarding the origin of baryon asymmetries $\mathcal{B}$ and $\mathcal{B}'$ in two sectors. As a simplest possibility, one can consider a (unspecified) baryogenesis mechanism acting separately in O and M worlds in identical manner [50]. Then two possible realizations of the discrete inter-sector symmetry, $\mathcal{Z}_2$ and $\mathcal{Z}_2^{LR}$, will have different implications for the relative sign

between $\mathcal{B}$ and $\mathcal{B}'$. Namely, in the case of $\mathcal{Z}_2^{LR}$ symmetry all B-violating processes between O particles must have the same CP-violation pattern as B'-violating ones in terms of M particles, so that $\mathcal{B}$ and $\mathcal{B}'$ must be of the same sign. As for the case of $\mathcal{Z}_2$ symmetry, CP-violating pattern becomes identical between the O particles and M antiparticles, and thus $\mathcal{B}$ and $\mathcal{B}'$ must have the opposite signs. Let us stress that we are talking about the relative sign; the absolute values of $\mathcal{B}$ and $\mathcal{B}'$ in the context of these mechanisms can be different since the out-of-equiilbrium conditions in the O and M worlds can be different due to the difference of their temperatures [50].

In the SM context, conservation of B and L is related to accidental global symmetries of the Lagrangian at the level of the renormalizable terms. However, these global symmetries can be broken by higher order operators emerging from some new physics at high energy scales. In particular, L should be violated if the neutrinos are the Majorana particles while the neutron-antineutron mixing would violate B. In both cases B − L will be violated, just what is needed for the baryogenesis.

Namely, the neutrino Majorana masses can be induced via the seesaw mechanism which involves heavy Majorana fermions $N$ coupled to the leptons $\ell_L = \ell$ via the Yukawa terms $YN\ell\phi + $ h.c., $Y$ being the matrix of the Yukawa coupling constants which are generically complex (the Lorentz and family indices are suppressed). The analogous Majorana fermions $N'$ and their Yukawa couplings $Y'N'\bar{\ell}'\bar{\phi}' + $ h.c., should be at work in M sector. Hence, by integrating out the heavy fermions, one obtains the dimension-5 effective operators

$$\frac{A}{M}(\ell\phi)^2 + \frac{A'}{M}(\bar{\ell}'\bar{\phi}')^2 \; + \; \text{h.c.,} \tag{3}$$

where $M$ is the mass scale of heavy Majorana fermions and $A$ and $A'$ are the symmetric matrices of dimensionless 'coupling' constants, for which $Z_2$ parity implies $A' = A$ whereas $Z_2^{LR}$ implies $A' = A^*$. These operators violate respectively L and L' by two units, and after inserting the VEVs $\langle\phi\rangle$ and $\langle\phi'\rangle$ they induce the Majorana masses of the O and M neutrinos. On the other hand, this mechanism suggests the leptogenesis mechanism which can induce baryon asymmetry via the decays of heavy Majorana fermions $N \to \ell\phi(\bar{\ell}\bar{\phi})$ which violate B − L and also CP (due to complex couplings $Y$). Then analogous decays $N' \to \ell'\phi'(\bar{\ell}'\bar{\phi}')$ should induce BA in the M sector. Thus, since $\mathcal{B}$ is known to be positive, $\mathcal{Z}_2$ symmetry $(\ell \leftrightarrow \bar{\ell}')$ implies negative $\mathcal{B}'$ while in the case of $\mathcal{Z}_2^{LR}$ $(\ell \leftrightarrow \ell')$ $\mathcal{B}'$ will be positive.

However, in the absence of cross-interaction terms $\mathcal{L}_{\text{mix}}$ (2), no experiment can discriminate between $\mathcal{B}' > 0$ and $\mathcal{B}' < 0$. For identifying the sign of $\mathcal{B}'$, one needs the lepton (or baryon) violating interactions between the O and M particles. Without such interactions, the question of the BA sign in mirror becomes metaphysical.

Such interactions can be naturally induced in the context of the above seesaw picture. In fact, there is no fundamental reason which can prevent some of $N$-fermions (namely, the SM singlets) to interact also with M leptons and vice versa, $N'$-ones to interact with O leptons. Then the Yukawa couplings, $\widetilde{Y}N'\ell_L\phi + \widetilde{Y}'N\bar{\ell}'\bar{\phi}' + $ h.c., can induce the 'mixed' effective operators,

$$\frac{\widetilde{A}}{M}(\bar{\ell}'\bar{\phi}')(\ell\phi) \; + \; \text{h.c.,} \tag{4}$$

which violate both L and L' by one unit, but conserves $\overline{L} = L + L'$. After inserting the Higgs VEVs, these operators induces the mixing between $\nu$ and $\bar{\nu}'$ states. If the constants $\widetilde{Y}, \widetilde{Y}'$ are much smaller than $Y, Y'$ and the new couplings do not bring two sectors in equilibrium, then the leptogenesis in O and M worlds remains dominated respectively by the decays $N \to \ell\phi$ and $N' \to \ell'\phi'$. Thus, for our BA being positive, $\mathcal{B} > 0$, for the mirror BA we still have $\mathcal{B}' < 0$ or $\mathcal{B}' > 0$, (depending on assumed symmetry $\mathcal{Z}_2$ or $\mathcal{Z}_2^{LR}$. Therefore, once ordinary stars during their burning produce the LH neutrinos $\nu$ (rather than RH antineutrinos $\bar{\nu}$), in the case of $\mathcal{B}' < 0$ the mirror stars should produce the LH states $\bar{\nu}'$ which oscillate into $\nu$. However, if $\mathcal{B}' > 0$, M stars will produce the RH states $\nu'$ which oscillate into our antineutrinos $\bar{\nu}$. (Notice, that in the absence of operators (3), with only $\overline{L}$-conserving operator (4) at work, neutrinos become the Dirac particles with their LH

components $\nu_L$ belonging to O sector and RH components $\nu_R'$ belonging to M sector. Of course, in this case $\nu - \bar{\nu}$ oscillations cannot take place.)

For example, the ordinary SN explosions are believed to be accompanied by the short *neutronization* burst of the *neutrinos* $\nu_e$. Their oscillation into sterile states $\bar{\nu}'$ can create some deficit in the expected flux. which deficit, however, can be difficult to quantify mainly due to uncertainties in the SN core-collapse modelling. On the other hand, the neutronization bursts from the mirror SN explosions, depending on the sign of $\mathcal{B}$, can be observed in terms of our *neutrinos* or *antineutrinos*. Namely, for $\mathcal{B} > 0$, these bursts will produce $\nu_e'$ which, via the oscillation $\nu_e' \to \bar{\nu}_e$, can be observed as the *antineutrino* bursts without the optical SN counterpart. Observation of such bursts can be a smoking gun signal which can shade the light on the mirror nature of the sterile neutrinos and of the DM in general.

However, the BA of opposite signs between the O and M sectors can be generated also in the case of $\mathcal{Z}_2^{LR}$ symmetry, if the couplings $\widetilde{Y}, \widetilde{Y}'$ are comparable to $Y, Y'$. In fact, one can simplify the above scenario and assume heavy O and M fermions are the same particles, $N \equiv N'$, in which case the Yukawa terms read as $YN\ell\phi + Y'N\bar{\ell}'\bar{\phi}' + \text{h.c.}$, and thus all operators (3) and (4) are induced at once by the exchange of $N$-fermions. In this case the co-leptogenesis in both sectors can take place by CP-violation in scattering processes as discussed in Refs. [81–84]. This scenario assumes that after inflation the O and M sectors are reheated asymmetrically, with $T \gg T'$, and masses $\mathcal{N}$ fermions between are larger than the reheating temperature. Nevertheless, the operators (4) mediate scattering processes as $\ell\phi \to \bar{\ell}'\bar{\phi}'(\ell'\phi')$, etc. which violate both L and L', and they are out-of-equilibrium. The invariance under $\mathcal{Z}_2^{LR}$ ($\ell \leftrightarrow \ell'$) for the Yukawa couplings implies $Y' = Y^*$, in which case the CP-violating factors in the above scattering processes are non-zero, and $\mathcal{B}$ and $\mathcal{B}'$ are induced with the opposite signs [39]. (Interestingly, this mechanism is ineffective in the case of $\mathcal{Z}_2$ symmetry yielding $Y' = Y$ since CP-violating factors appear to be vanishing [39].) Let us remark that this mechanism implies $\Omega_{b'} > \Omega_b$, which is related to the fact that M sector is colder and the produced $B' - L'$ suffers less damping [84]. Hence, it can naturally explain the observed cosmological fractions of the baryons and DM, $\Omega_{b'}/\Omega_b \simeq 5$, which also makes clear who has eaten more oysters, the Walrus or the Carpenter.

Let us turn now to $n - n'$ mixing (1). It can be originated from the following effective interactions:

$$\frac{1}{\mathcal{M}^5}(\bar{u}'\bar{d}'\bar{d}')(udd) + \text{h.c.}, \tag{5}$$

which conserve $\bar{B} = B + B'$ [5]. Here $\mathcal{M}$ is some large scale of underlying new physics, and the parentheses contain the gauge invariant spin $1/2$ combinations of ordinary quarks $u, d$ and mirror antiquarks $\bar{u}', \bar{d}'$ (the gauge and Lorentz indices are omitted). So, $nn'$ mixing mass $\varepsilon$ in (1) can be estimated as:

$$\varepsilon = \frac{C^2 \Lambda_{\text{QCD}}^6}{\mathcal{M}^5} \simeq C^2 \left(\frac{10\,\text{TeV}}{\mathcal{M}}\right)^5 \times 10^{-15}\,\text{eV}, \tag{6}$$

where $C = O(1)$ is the operator dependent numerical factor in the determination of the matrix element $\langle 0|udd|n\rangle$. Although we single out the possibility of $n - n'$ mixing, generically all neutral O and M baryons can have analogous mixings as e.g., $\Lambda - \Lambda'$ due to operators similar to (5) involving $uds$ and $u'd's'$.

In the context of UV complete renormalizable theory, the operators (5) can be induced via a seesaw like mechanism involving new heavy particles, as color-triplet scalars $S$ and $S'$ and a neutral Dirac fermion $F$, so that we have $\mathcal{M}^5 \sim M_S^4 M_F$ modulo the Yukawa coupling constants [5,9]. Hence, for color scalars at few TeV, the underlying theories can be testable at the LHC and future accelerators [9]. Interestingly, if $F$ fermions are allowed to also have a small Majorana mass term, $\mu \ll M_F$, then the same seesaw mechanism would induce also $\Delta B = 2$ operators $\sim (udd)_R^2$ (and their M counterparts) leading to the neutron-antineutron ($n - \bar{n}$) with mixing $\varepsilon_{n\bar{n}} = (\mu/M_F)\varepsilon$ [5,9]. Thus, the origin of both $n - n'$ and $n - \bar{n}$ oscillation phenomena can be related to the same new physics. In addition,

$n - \bar{n}'$ and $\bar{n} - n'$ mixings can also be induced via dimension 10 effective operators [85]. These operators can also induce BAs in both sector via CP-violation of scattering processes $dS \rightarrow d'S'$ etc. and, depending on the Yukawa phases, $\mathcal{B}$ and $\mathcal{B}'$ can have the or the opposite signs.

Clearly, the neutron disappearance experiments cannot identify the sign of $\mathcal{B}'$, and moreover, they cannot distinguish between $n \rightarrow n'$ and $n \rightarrow \bar{n}'$ transformations. In the presence of only $nn'$ mixing which conserves $\widetilde{B} = B + B'$, neither the regeneration experiments $n \rightarrow n' \rightarrow n'$ are sensitive to the sign of $\mathcal{B}'$. (Notice however, that if both $nn'$ and $n\bar{n}'$ mixings are present and so $\widetilde{B}$ is not conserved, then the effect of the antineutron regeneration $n \rightarrow n'\bar{n}' \rightarrow \bar{n}$ can be observed in the experiments [85]. However, here we shall not concentrate on this possibility and discuss only the case of $nn'$ mixing which conserves $\widetilde{B}$.

However, the sign of $\mathcal{B}'$ can be directly tested by the mirror neutron transformation to our neutron, $n' \rightarrow n$. For example, if $\mathcal{B}'$ is negative, then $\bar{n}'$ states bound in the DM component represented by the mirror nuclei. (As it was shown in [50], mirror matter should be dominated by ${}^4\text{He}'$ component, and perhaps some heavier nuclei produced by the evolution of mirror stars [53].) will have, due to mixing (1), an $\bar{n}$ admixture $\theta \sim (\varepsilon/\Delta E)$, where $\Delta E$ is the binding energy per nucleon of few MeV. Therefore, when the mirror nuclei pass the Earth, these $\bar{n}'$ can annihilate with the ordinary nucleons in the DM detectors with the cross-section $\propto \theta^2$ producing pions. However, in reality this is rather a gedanken possibility since the probability of such annihilations is very small: e.g., for $\varepsilon \sim 10^{-15}$ eV we have $\theta^2 \sim 10^{-43}$. Hence, even for large statistics, the question is how to the discriminate these events from the background. Another possibility is related to the mirror cosmic rays. Namely, $\bar{n}'$, produced by spallation of M nuclei in the mirror photon background or by scattering on interstellar mirror gas, can oscillate into our antineutrons $\bar{n}$ which then decay as $\bar{n} \rightarrow \bar{p}e^+\nu$ producing the cosmic antiprotons and positrons.

However, most interesting is the effect of $\bar{n}' \rightarrow \bar{n}$ transition on mirror (anti)neutron stars which can produce ordinary antimatter cores in their interiors. In the following we shall concentrate on this effect. In next section, for warming up, we briefly discuss the effects of $n \rightarrow n'$ transition in ordinary NS and the limits on the characteristic time of this transition.

## 4. $n - n'$ Transition in Neutron Stars

Neutron stars are presumably born after the supernova (SN) explosions of massive progenitor stars followed by the neutronization of their iron cores. The NS are formed with very high rotation speed and with the surface magnetic fields as large as $10^8 \div 10^{15}$ Gauss. This makes many of them observable as pulsars due to their electromagnetic radiation. The structure of NS is onion-like: the inner core of dense nuclear liquid dominantly consisting of Fermi-degenerate neutrons (with about 10% fraction of protons and electrons, and perhaps of heavier baryons and muons), the inner crust dominated by the heavy nuclei, and the rigid outer crust at the surface (for reviews, see e.g., [86–88]). The NS mass-radius ($M$–$R$) relations depend on the equation of state (EoS), i.e., the pressure–density relation in dense nuclear matter. The masses of the known pulsars range within $M = (1 \div 2) M_\odot$, and the observations of $2M_\odot$ ones disfavor the too soft EoS. Some of the realistic EoS reviewed in Refs. [88,89] can afford the NS masses up to $(2.0 \div 2.5) M_\odot$ and predict the NS radii in the range $R \simeq (10 \div 14)$ km.

The number of baryons $N$ in the NS is related to its mass $M$ in approximately linear way, $N = \kappa(M/n)$, where $m$ is the nucleon mass and the EoS dependent factor $\kappa$ accounts for the gravitational binding. The deficit between the gravitational mass $M$ and equivalent baryonic mass $M_b = mN$, $M_b - M = (\kappa - 1)M$, corresponds to the gravitational binding energy. Hence, we have:

$$N = \kappa(M/m) \approx \left(\frac{M}{1.5 \, M_\odot}\right) \times 2 \cdot 10^{57}, \tag{7}$$

where we take into account that for the typical NS masses $M \simeq 1.5\,M_\odot$ the realistic EoS imply $\kappa = 1.1$ or so.

The core-collapse of an O star should produce an NS practically consisting of ordinary nuclear matter. (Though some tiny amount of M matter can be accreted by the progenitor star during its lifetime, or produced via $n - n'$ oscillation of the neutrons involved in certain chains of nuclear reactions at late stages of its evolution before the core-collapse.) However, once the NS is formed, then $n \to n'$ transitions in its liquid core can effectively produce M neutrons, thus transforming the initial NS into a mixed star consisting in part of M matter. This process can be described by the Boltzmann equations,

$$\frac{dN}{dt} = -\Gamma N + \Gamma' N', \qquad \frac{dN'}{dt} = \Gamma N - \Gamma' N', \tag{8}$$

where $N(t)$ and $N'(t)$ respectively are the amounts of O and M baryons in the star at the time $t$, $\Gamma$ is the rate of $n \to n'$ conversion and $\Gamma'$ is the rate of the inverse process $n' \to n$. Starting from a newborn NS, with $N = N_0$ and $N' = 0$ at $t = 0$, then $N'(t)$ will increase while $N'(t)$ will decrease in time. However, the overall amount of baryons remains constant, $N(t) + N'(t) = N_0$, since $n - n'$ oscillation conserves the combined baryon number $\overline{B} = B + B'$. Since the inverse reaction rate $\Gamma'$ is in fact negligible, Equation (8) reduce to a single equation:

$$\frac{dX}{dt} = \Gamma\,(1 - X)\,, \tag{9}$$

where $X(t) = N'(t)/N_0$ is the fraction of M baryons at the time $t$ and $N(t)/N_0 = 1 - X(t)$. The transition rate $\Gamma = \Gamma(X)$ is not constant since it depends on the NS composition which itself evolves in time, and it can be presented as:

$$\Gamma(X) = \Gamma(0)\mathcal{F}(X), \tag{10}$$

where $\Gamma(0) = 1/\tau_\varepsilon$ is the 'starting' rate of conversion for a given star (at $t = 0$, i.e., $X = 0$) with $\tau_\varepsilon$ being the characteristic transition time (which depends on $nn'$ mixing strength as $\tau_\varepsilon \propto 1/\varepsilon^2$), and the function $\mathcal{F}(X)$ (normalized as $\mathcal{F}(0) = 1$) comprises the dependence on the mirror fraction $X$ in the star. By integrating this equation, we obtain the time $t$ at which the M fraction in the star reaches the value $X$:

$$t(X) = \int_0^X \frac{dx}{\Gamma(x)(1 - x)} = \tau_\varepsilon \int_0^X \frac{dx}{(1 - x)\mathcal{F}(x)}. \tag{11}$$

Thus, for $X \ll 1$ we have:

$$\dot{X} = 1/\tau_\varepsilon \qquad \longrightarrow \qquad N'/N_0 = t/\tau_\varepsilon, \tag{12}$$

meaning that for $t \ll \tau_\varepsilon$ the fraction $X$ increases linearly with time. However, with $X$ growing the evolution gradually slows down since $\mathcal{F}(X)$ is a decreasing function of $X$, and $\mathcal{F}(X) \to 0$ in the limit $X \to 1/2$ when the star reaches a maximally mixed configuration with equal amounts of O and M baryons in its interior.

Let us evaluate now the transition time $\tau_\varepsilon$ supposing that it is much larger than the NS cooling time. Then the star can be considered as a degenerate nuclear medium, containing the neutrons and protons respectively with the fractions $x_n \approx 0.9$ and $x_p \approx 1 - x_n \approx 0.1$, in which the evolution of $n - n'$ system is described by effective Hamiltonian:

$$H = \begin{pmatrix} E & \varepsilon \\ \varepsilon & E' \end{pmatrix}, \tag{13}$$

where $E$ and $E'$ are the effective energies of $n$ and $n'$ at a given momentum $p$, and the off-diagonal term $\varepsilon$ is induced by $n - n'$ mass mixing (1). For the neutron the free particle dispersion relation $E(p) = (m^2 + p^2)^{1/2}$ is no more applicable since it is affected by the

strong repulsive and attractive interactions in the dense matter. Namely, these interactions (mediated respectively by the vector and scalar mesons) modify the dispersion relation as $E(p) = (p^2 + m_*^2)^{1/2} + V$, where $m_* = m - S$. This means that the neutrons propagate in the background of external scalar and vector fields which values are determined by the mean scalar and vector densities of baryons $N = n, p$. The values of $S$ and $V$ depend on the nuclear interaction model and respective EoS for the nuclear matter, and for a given model they are determined by the baryon density $n_b$. For example, contributions of the neutron component to $V$ and $S$ are proportional respectively to $\langle \bar{n}\gamma^0 n \rangle = x_n n_b$ and $\langle \bar{n}n \rangle$, where the latter value depends on $n_b$ in more complicated way. At supra-nuclear densities both of these values are $\sim 100$ MeV, with the vector interactions being more effective, $V > S$. As for mirror neutrons, in the young NS their density is negligible, $n_b' \ll n_b$, and we can take $E'(p) = (m^2 + p^2)^{1/2}$. (For simplicity, we consider that $n$ and $n'$ are exactly degenerate in mass, $m' = m$, though in our discussions will be applicable as well in the case of small mass splitting, $\Delta m = |m - m'| < 1$ MeV or so.) Therefore, the medium induced energy splitting $\Delta E = E - E'$ is pretty large. Namely, in non-relativistic limit $p^2/m_*^2 \ll 1$ we have $\Delta E \approx V - S \sim (10 \div 100)$ MeV. Hence, a small splitting $\Delta m \ll 1$ MeV between $n - n'$ masses can be neglected as well as the Zeeman energy $|\mu B|$ induced by the neutron magnetic moment $\mu$ since it is $< 10^{-2}$ MeV even in magnetars with the magnetic field $B \sim 10^{15}$ G.

The Hamiltonian eigenstates are:

$$n_1 = cn - sn', \quad n_2 = sn + cn', \tag{14}$$

where $c = \cos\theta$ and $s = \sin\theta$, with the mixing angle given by $\tan 2\theta = 2\varepsilon/\Delta E$. Since $\Delta E \gg \varepsilon$, the mixing angle is extremely small:

$$\theta \approx \frac{\varepsilon}{\Delta E} = \varepsilon_{15} \left( \frac{100 \text{ MeV}}{\Delta E} \right) \times 10^{-23}, \tag{15}$$

where for convenience we take $n - n'$ mixing mass in units of $10^{-15}$ eV, i.e., $\varepsilon = \varepsilon_{15} \times 10^{-15}$ eV. So, the mixing angle is extremely small so that one can set $c = 1$ and $s = \theta$.

While the ordinary and mirror neutrons, $n$ and $n'$, can be considered as the 'flavor' eigenstates having separate (respectively O and M) strong interactions, the non-diagonal interactions emerge in the basis of Hamiltonian eigenstates. Namely, the neutron interactions with a target nucleon $N = n, p$ described by the effective operators $(\bar{n}\gamma n)(\bar{N}\gamma N)$, where $\gamma = 1, \gamma^5, \gamma^\mu$ etc. stand for possible Lorentz structures (the coupling constants are omitted), in terms of the Hamiltonian eigenstates (14) read:

$$(c^2 \overline{n_1}\gamma n_1 + cs\, \overline{n_1}\gamma n_2 + cs\, \overline{n_2}\gamma n_1 + s^2 \overline{n_2}\gamma n_2)(\bar{N}\gamma N), \tag{16}$$

which contain the mixed terms between $n_1$ and $n_2$.

The mirror neutron product rate can be calculated along the lines discussed in Ref. [31]. We can consider that the initial NS consists of the degenerate Fermi liquid dominated by ordinary neutrons $n \approx n_1$. The processes $n_1 N \to n_1 N$, $N = n, p$, are Pauli-blocked, but for transitions $n_1 N \to n_2 N$ the Pauli blocking has only a partial role. (In the following notations we take $n_1 \approx n$ and $n_2 \approx n'$ and denote these processes as $nN \to n'N$.) For example, the scattering $np \to n'p$ will have an amplitude $\theta f_{np}$, with $f_{np}$ being the neutron-proton scattering amplitude. Therefore, cross section of this process will be $\theta^2 \eta_p \sigma_{np}$ where $\sigma_{np}$ is $np \to np$ elastic scattering cross section and $\eta_p$ is the Pauli blocking factor which takes into account the phase space restrictions, namely that final $p$ should have the momentum larger than the proton Fermi momentum in the degenerate medium. Analogously, for $nn \to n'n$ process one has to consider the Pauli factor $\eta_n \approx 0.18$ which takes into account that the momentum of final $n$ should be larger than the neutron Fermi momentum $p_F = (3\pi^2 x_n n_b)^{1/3}$.

Then the rate of $n - n'$ conversion in the medium with the baryon density $n_b$ can be estimated as:

$$\Gamma(n_b) = \theta^2 \sigma_{\text{eff}} \, v \, n_b \simeq \frac{3 \sigma_{\text{eff}} \, n_b^{4/3}}{\Delta E^2} \frac{\varepsilon^2}{m} = A(n_b) \frac{\varepsilon^2}{m},$$

$$A(n_b) \simeq 10^2 \left( \frac{\sigma_{\text{eff}}(n_b)}{50 \, \text{mb}} \right) \left( \frac{30 \, \text{MeV}}{\Delta E(n_b)} \right)^2 \left( \frac{n_b}{0.3 \, \text{fm}^{-3}} \right)^{4/3}, \tag{17}$$

where we denote: $\sigma_{\text{eff}} = \eta_n x_n \sigma_{nn} + \eta_p x_p \sigma_{np}$, $x_n$ and $x_p = 1 - x_n$ being respectively the neutron and proton fractions (for simplicity, we neglect the smaller fraction of heavier baryons). In addition, take the mean scattering velocity as $v \simeq p_F / m$, and use Equation (15) for mixing angle $\theta$. The dimensionless parameter $A(n_b)$ depends on nuclear interaction model, and in the context of a given model its value is fully determined by the baryon density $n_b$ which varies by an order of magnitude from the NS centre to the outer edge of its liquid core. Notice that $\sigma_{\text{eff}}$ for a lab system momenta $p \sim p_F \propto n_b^{1/3}$ is a decreasing function of $p \propto n_b^{1/3}$ whereas $\Delta E(n_b)$ is an increasing function of $n_b$ with a non-trivial shape. Therefore, $A(n_b)$ appears to be a rather mild function of $n_b$. Namely, in Equation (17) it was estimated by taking typical baryon densities $n_b \simeq 0.3 \, \text{fm}^{-3}$ and respective Fermi momenta $p_F \simeq 400 \, \text{MeV}$ at which one has $\sigma_{\text{eff}} \simeq 50 \, \text{mb}$ or so, as one can estimate using the experimental values for the cross sections $\sigma_{nn}$ and $\sigma_{np}$ at $p_{\text{lab}} \simeq p_F$.

In this way, the effective 'starting' rate of the NS transformation is given by,

$$\Gamma(0) = A \frac{\varepsilon^2}{m} = A \, \varepsilon_{15}^2 \times 10^{-48} \, \text{GeV}, \tag{18}$$

where $A = \langle A(n_b) \rangle_{\text{NS}}$, with the parentheses meaning the average over the baryon density profile in the NS. Once again, $\Gamma(0)$ depends on the nuclear interaction model which in turn determines the EoS of dense nuclear matter, and in the context of given model it will be determined by the profiles of the baryon density $n_b$ in the NS. In fact, this value mildly depends on the overall amount of baryons, which is related to the NS mass via Equation (7), and roughly it can vary around $A \sim 10^2$.

Thus, the effective time of $n - n'$ transformation of the NS can be estimated as:

$$\tau_\varepsilon = 1/\Gamma(0)^{-1} \sim \varepsilon_{15}^{-2} \times 10^{22} \, \text{s} \tag{19}$$

and so for $\varepsilon \sim 10^{-15} \, \text{eV}$ we expect $\tau_\varepsilon \sim 10^{15} \, \text{yr}$ or so. Once again, this is just an order of magnitude estimation, and in the following we use as a parameter directly the transition time $\tau_\varepsilon$.

Let us discuss now the fate of produced mirror neutrons. Due to the Pauli blocking, the process $nn \to nn'$ process takes place at the neutron momenta close to the Fermi surface, so that $n'$ is typically produced with the energy $E_{n'}(n_b) = p_F^2/2m \simeq (n_b/0.3 \, \text{fm}^{-3})^{2/3} \times 100 \, \text{MeV}$. Hence, in the case of $t \ll \tau_\varepsilon$, the rate of energy production in the NS can be obtained by integrating the product of transition rate $\Gamma(n_b)$ (17) and the typical energy $E_{n'}$ over the NS volume. Thus, taking into account Equation (7) for the total amount baryons in the star, the energy production rate via $n - n'$ transition can be estimated as:

$$\dot{\mathcal{E}}_{nn'} \simeq \frac{\langle E_{n'} \rangle_{\text{NS}} N}{\tau_\varepsilon} \simeq \left( \frac{M}{1.5 \, M_\odot} \right) \times \frac{10^{31} \, \text{erg/s}}{\tau_{15}}, \tag{20}$$

where $\tau_{15} = (\tau_\varepsilon/10^{15} \, \text{yr})$ is the NS transition time in units of $10^{15} \, \text{yr}$.

The produced energy should be radiated away by the cooling of hot mirror material, in terms of mirror photons and neutrinos. In fact, mirror neutrons produced via $n - n'$ transitions at initial stage of the NS evolution will decay as $n' \to p'e'\bar{\nu}'$ producing a hot plasma of of M protons and electrons gravitationally trapped in the NS interior, and cooling process will be dominated by mirror neutrino emission. Then the nucleosynthesis processes

involving $n'$ and $p'$ will be ignited, which will produce heavier mirror nuclei. If by time the density of mirror core in the NS will reach a sufficiently large value, the "neutronization" will occur and the core of liquid M neutrons will be formed. Needless to say, M cores in the ordinary NS can be detectable for a M observer via the mirror photon emission in the far UV and X ray ranges.

On the other hand, $n - n'$ transition should heat up also the ordinary nuclear matter in the NS. Namely, the neutron disappearance in the reaction $nn \to nn'$ leaves the empty level in the Fermi see which will be filled by the neutron transition for the higher levels. Once again, as far as reactions $nn \to nn'$ can take place only close to the Fermi surface, the energies corresponding to the latter transitions should be an order of magnitude smaller than the typical energy $E_{n'}(n_b) = p_F^2/2m$ of produced $n'$ states. Then, by equating the energy production rate by this transition as $\dot{\mathcal{E}}_{nn} \sim 0.1 \, \dot{\mathcal{E}}_{nn'}$ or so, the NS surface temperature can be estimated as [31]:

$$T \simeq \left( \frac{M}{1.5 \, M_\odot} \right)^{5/12} \times \frac{1.5 \cdot 10^5 \text{ K}}{\tau_{15}^{1/4}}, \tag{21}$$

simply by equating $\dot{\mathcal{E}}_{nn} = 4\pi R^2 \cdot \sigma T^4$ where $\sigma$ is the Stefan–Boltzmann constant and the NS radius is taken as $R \simeq 12$ km.

The standard cooling mechanisms predict a sharp drop of the surface temperatures with the age of the pulsar. Namely, one would expect $T \simeq 10^4$ K after $10^7$ yr and $T \simeq 10^3$ K after $10^8$ yr. On the other, the observations of some old pulsars as PSR B0950+08, J0437−4715 and J2123−3358 (with characteristic ages $\tau_c$ from few $\times 10^7$ yr to few Gyr) show that they are still warm, with surface temperatures $T \simeq (1 \div 3) \times 10^5$ K [90–92]. This means that some heating mechanism should operate it the NS which, according to estimation (21), can in fact be provided by $n - n'$ transition process if the effective transition time is $\tau_\varepsilon \simeq 10^{15}$ yr or so.

There is an interesting exclusion. Namely, a slow pulsar J2144−3933 with $\tau_c = 3.3$ Gyr is rather cold: its observations set only an upper limit $T < 4.2 \times 10^4$ K [93] which would be compatible with (21) if $\tau_\varepsilon > 10^{17}$ yr or so. However, the latter bound can be avoided if we assume that this pulsar, with the unknown mass, is a quark star for which the above heating mechanism becomes ineffective.

In fact, the compact astrophysical objects could exist also in the form of hybrid stars with the quark matter core, or quark stars entirely consisting of quark matter (for a review, see e.g., [94,95]). In particular, according to the Bodmer–Witten hypothesis [96,97], strange quark matter can be energetically favored state at very large densities. In this way, the NS consisting of nuclear matter can be metastable states which can be transformed into strange quark stars (QS). This phase transition can occur once the NS mass increases a certain critical value, e.g., by accretion from the companion in neutron star X-ray binaries [98] or by the matter fall-back shortly after the SN explosion [99]. which can also induce a powerful gamma-ray burst.

This picture implies that the heavier compact stars, and in particular the ones with masses $\simeq 2M_\odot$, should be the QS rather than the NS. In fact, the quark matter EoS can be stiff enough to afford the QS with rather large masses. Recently, such NS→QS transitions were studied for the different EoS of nuclear matter in Ref. [100] which shows that all compact (non BH) objects with masses larger than $1.6 \, M_\odot$ or so in fact can be the QS.

While $n - n'$ transition is possible in the NS where the neutrons are bound by gravity, it should be ineffective in the QS consisting of strange quark matter which under the Bodmer–Witten hypothesis should be self-bound which makes (like in the nuclei) the production of $n'$ energetically disfavoured.

As was mentioned in the introduction, in the NS, where the neutrons are bound by gravity, $n - n'$ transition is energetically convenient and it should gradually transform the NS into a mixed star consisting partially of mirror matter. On the other hand, in the QS $n - n'$ conversion should be ineffective, not only because of the absence of the neutrons in

the quark matter (multi-quark processes as $udd \rightarrow n'$ could work instead), but principally because of the self-boundness of quark matter which makes (like in the nuclei) this transition energetically disfavoured.

In the following, we shall concentrate on the NS, and conservatively take the lower limit on $n - n'$ transition time as $\tau_\varepsilon > 10^{15}$ yr or so.

## 5. Antimatter Cores in Mirror Neutron Stars

Let us reverse now the situation and consider neutron stars in the M sector. As we have anticipated in the introduction, the BA $\mathcal{B}'$ in M world can be positive or negative, depending on the baryogenesis mechanism. Hence, if $\mathcal{B}' > 0$ all compact mirror stars should be the MNS, whereas if $\mathcal{B}' < 0$ they all should be the AMNS. As far as the O and M sectors have the same microphysics by mirror parity, the EoS describing the O and M nuclear matters should be identical, and, needless to say, it should be identical for the nuclear and antinuclear matters by C invariance of strong interactions. In other words, the NS, MNS and AMNS should have the same $M$–$R$ relations.

In the absence on $n - n'$ mixing these will be dark compact objects, sort of solar mass MACHOs which can be detected by microlensing, but ordinary observer cannot distinguish between the MNS and AMNS. If $n - n'$ mixing is switched on, then $n' \rightarrow n$ transitions will create O matter in the MNS interior, while in the AMNS $\bar{n}' \rightarrow \bar{n}$ will take place forming the O antimatter. These transitions with the effective time and energy production rate given again by Equations (19) and (20), would form hot cores which can be visible for us as bright compact stars with high temperatures. Clearly, with this electromagnetic emission one cannot distinguish whether the core composed of O matter or O antimatter. However, the two cases can be discriminated by the accretion of ordinary gas which, in the case of the AMNS, will annihilate with the antibaryons. Thus, detection $\gamma$-ray sources with a typical baryon–antibaryon annihilation spectrum [43] can be the way to determine the BA sign in mirror world.

Let us consider an AMNS of a typical mass $M \simeq 1.5\,M_\odot$. Transitions $\bar{n}' \rightarrow \bar{n}$ will produce O antimatter in its liquid core. The production rate (antibaryon per second), can be estimated according to Equation (12):

$$\dot{N}_{\bar{b}} = N_0'/\tau_\varepsilon \approx \frac{1}{\tau_{15}} \left( \frac{M}{1.5\,M_\odot} \right) \times 6 \cdot 10^{34}\ \text{s}^{-1}, \tag{22}$$

where $N_0'$ is the initial amount of the AMB in star which is related to its mass via Equation (7). Thus, the fraction of ordinary antibaryons produced in the AMNS during the time $t$ is $X = N_{\bar{b}}/N_0 = t/\tau_\varepsilon$. For a star with $M = 1.5\,M_\odot$ and age $t = 5$ Gyr this would give the antibaryon amount $N_{\bar{b}} \approx 10^{52}/\tau_{15}$, equivalent to a few times the Earth's mass.

In this way, antimatter core will be formed in the AMNS interior. In fact, antineutrons $\bar{n}$ produced via $\bar{n}' \rightarrow \bar{n}$ transition at the initial stages will undergo $\beta$-decay $\bar{n} \rightarrow \bar{p}e^+\nu_e$ forming a hot plasma consisting of antiprotons and positrons. After few years, when the density of produced antimatter reach $10^{26}/\text{cm}^3$ or so, nuclear reactions between $\bar{p}$ and $\bar{n}$ will become effective and start to form antinuclei. All energy produced at these stages will be emitted in terns of the neutrinos and photons, with the rate given by Equation (20). Hence, the AMNS can be seen as the photon sources in the far UV or soft X ray ranges.

On the other hand, the AMNS will accrete O gas while it travels in diffuse interstellar medium (ISM). According to Bondi-Hoyle-Littleton formula, the accretion rate (baryon per second) reads:

$$\begin{aligned} \dot{N}_b &= (2GM)^2\, v^{-3}\, n_{\text{ism}} \\ &\simeq \frac{1}{v_{100}^3} \left( \frac{M}{1.5\,M_\odot} \right)^2 \left( \frac{n_{\text{ism}}}{1/\text{cm}^3} \right) \times 10^{32}\ \text{s}^{-1}, \end{aligned} \tag{23}$$

where $n_{\text{ism}}$ is the ISM number density and $v = v_{100} \times 100$ km/s is the star velocity relative to the ISM.

Hence, if the transition time is in the range $\tau_\varepsilon = (10^{15} \div 10^{17}$ yr for the AMNS with $v > 100$ km/s the antibaryon production rate (22) is much larger than the baryon accretion rate (23), $\dot{N}_{\bar{b}} \gg \dot{N}_b$. (The pulsar kick velocities are typically of about 100 km/s and some achieve even 1000 km/s.) In this case, the antimatter core can be formed while the accreted baryons will annihilate on its surface. The annihilation photons will be produced with the rate $L_\gamma = l_\gamma \dot{N}_b$, where $l_\gamma \approx 4$ is the average multiplicity of $\gamma$s per $p\bar{p}$ annihilation [43]. The rate of energy production is $2m\dot{N}_b$, about a half of which will be emitted from the core surface surface in $\gamma$-rays. and another half will contribute to heating the core (in addition to (20)). Hence, the energy flux from such a source at a distance $d$ will be $J \simeq m\dot{N}_b/4\pi d^2$, or numerically

$$J \simeq \frac{10^{-12}}{v_{100}^3}\left(\frac{n_{\rm ism}}{1/{\rm cm}^3}\right)\left(\frac{M}{1.5\,M_\odot}\right)^2\left(\frac{50\,{\rm pc}}{d}\right)^2\frac{\rm erg}{{\rm cm}^2{\rm s}} \tag{24}$$

For the AMNS travelling with $v > 100$ km/s in the Milky Way (MW) this emission can be below the diffuse $\gamma$-background and the source may remain unresolved unless this source is closer than 50 pc or so. However, if the AMNS has less velocity, say $v \simeq 30$ km/s, and/or it incidentally crosses a high density region, such as, e.g., cold molecular cloud with $n_{\rm is} > 10^3/{\rm cm}^3$, the observability distance can be increased up to several kpc.

On the other hand, for the slow AMNS moving in galactic discs with $v < 10$ km/s, the antibaryons produced in its interior can be outnumbered by the accreted baryons, i.e., $\dot{N}_{\bar{b}} < \dot{N}_b$. In this case the antimatter core cannot be formed. since the produced antibaryons will readily annihilate with the already accreted baryons and the $\gamma$-ray luminosity will be $L_\gamma = l_\gamma \dot{N}_{\bar{b}}$. In other words, the $\gamma$-luminosity of the object will be defined by the lesser value between $\dot{N}_{\bar{b}}$ and $\dot{N}_b$.

The search for $\gamma$-ray sources with a spectrum compatible with baryon–antibaryon annihilation was recently performed in Ref. [41]. Analyzing 5787 sources included in the 4FGL catalog [42] based on 10 years of observations with the Fermi LAT, 14 candidates were found which were selected by applying the following criteria:

(i) extended candidates were excluded (with angular size larger than the LAT resolution at energies $E < 1$ GeV);

(ii) sources associated with objects known from other wavelengths and belonging to established $\gamma$-ray sources were excluded, as e.g., pulsars and active galactic nuclei;

(iii) sources with significant higher energy tail above a GeV were excluded since the baryon–antibaryon annihilation $\gamma$-spectrum should end up at the nucleon mass.

Interestingly, the distribution of the sources in the sky shown in Figure 1 of Ref. [41] very much resembles the distribution of observed pulsars. Only two candidates have galactic coordinates compatible with the MW disc, while the 11 candidates having galactic latitudes $|b| > 10°$ (among which seven candidates with $|b| > 30°$) can be assigned to the MW halo. Interestingly, the sources belonging to the disc are the brightest, with the energy fluxes $J \geq 10^{-11}$ erg cm$^{-2}$ s$^{-1}$, while the ones with higher galactic latitudes become increasingly fading, and the source J2330-2445 ($b = -71, 7°$) is the faintest, with $J < 2 \times 10^{-12}$ erg cm$^{-2}$ s$^{-1}$. In view of Equation (23), this may well explained by correlation of the accretion rate with the distribution of the ISM densities, though the velocity distribution of the stars remains the key issue.

It is probably too early to claim the discovery. These sources are all faint, close to the Fermi LAT detectability threshold, and they may well belong to a known $\gamma$-ray source classes, or maybe mimicked by imperfections of the background interstellar emission. In fact, the authors of Ref. [41] take a conservative attitude and translate their findings into an upper limit on the local fraction of such objects with respect to normal stars. The clear identification of these sources is very challenging, and requires a serious multiwavelength search.

The possibility of mirror neutron stars being the engines of our antimatter can have interesting implications since they can produce antinuclei in the ISM. Namely, for $\tau_\varepsilon \sim 10^{15}$ yr, transition $\bar{n}' \to \bar{n}$ in the AMNS produces about $10^{52}$ antibaryons forming the hot and dense

core in which nuclear reactions should form the antinuclei. However, these antinuclei are gravitationally trapped and the question is how they can escape from the star. This possibility can be provided by the mirror neutron star mergers. In the coalescence of the two AMNS, their small antimatter cores do not merge at the same instant but continue the orbiting, and then promptly explode due to the decompression producing a hot cloud of the neutron rich antinuclei. Most of these antinuclei, being stable only at the extreme densities, will decay into the lighter ones which are stable in normal conditions. Therefore, the antinuclei produced by such "sling" effect can leave the coalescence site and propagate in the outer space. In addition, they can have some additional acceleration if reasonably large magnetic fields are formed in the rotating ordinary anti-core during its evolution before the merger. The AMS-02 experiment hunting for the antinuclei in the cosmic rays has reported, as the preliminary results, the detection of eight antihelium events (among which two are compatible with antihelium-4 and the rest with antihelium-3), The fraction $\sim 10^{-8}$ of antihelium with respect to measured fluxes of the helium is too high to be explained by the conventional production mechanisms. Interestingly, the rate of the NS mergers $\sim 10^3 \, \text{Gpc}^{-3} \, \text{yr}^{-1}$, with $\sim 10^{52}$ antibaryons produced per a merger, is nicely compatible with this fraction of antihelium. In addition, our mechanism should produce also the heavier antinuclei, and thus AMS-02 can be the place where to find fantastic animals as anticarbon or antioxygen detection of which can become a key for many mysteries.

## 6. Discussion and Outlook

To conclude, we have discussed the possibility of M world having a negative BA, so that the M neutron stars are the AMNS, and $\bar{n}' - \bar{n}$ transition in their interior can create antimatter cores. The ordinary gas accreted from the ISM annihilating on the surface of these cores give rise to $\gamma$-rays with the typical spectrum of the baryon–antibaryon annihilation.

We have discussed these effects in the minimal situation, assuming that $n - n'$ mixing occurs only due to mass mixing (1), $n - \bar{n}'$ mixing is absent, and $n$ and $n'$ are exactly degenerate in mass. Under these circumstances, the experimental limits on mixing mass, $\varepsilon < 10^{-15}$ eV or so, imply that the NS transition process into mixed star is very slow, and the effective transition time $\tau_\varepsilon$ exceeds the universe age by several orders of magnitude [5]. In this case $n - n'$ transition should be an ongoing process in the existing neutron stars (or M neutron stars), and the limits on the surface temperatures of old pulsars imply $\tau_\varepsilon > 10^{15}$ yr or so [31].

However, our concept, in more general frameworks, permits some variations which we briefly mention below:

*Transitional magnetic moment.* In difference from the $n - \bar{n}$ system where transitional magnetic moment between $n$ and $\bar{n}$ is forbidden by Lorentz invariance, non-diagonal magnetic moment $\mu_{nn'}$ (or dipole electric moment) is allowed between $n$ and $n'$ [101–103] and they can be effectively induced in certain models of $n - n'$ mixing [30]. In this case the $n - n'$ transition time will depend on the magnetic field in the NS, and it can be simply estimated by replacing $\varepsilon$ into $|\mu_{nn'}B|$ in Equation (19) or, more concretely,

$$\varepsilon_{15} \longrightarrow \varepsilon_{15}^B = \left( \frac{\mu_{nn'}}{10^{-27} \, \text{eV/G}} \right) \left( \frac{B}{10^{12} \, G} \right). \tag{25}$$

Taking e.g., $\mu_{nn'} \sim 10^{-27}$ eV/G, which is 16 orders of magnitude smaller than the neutron magnetic moment itself, and making replacement (25) in Equation (22), we see that for a mirror magnetar ($B \sim 10^{15}$ G) the antimatter production rate will be $\sim 10^{40}/$s while for an old recycled AMNS with $B \sim 10^8$ G it will be $\sim 10^{26}/$s. Thus, the former objects should be very bright while the latter can be practically invisible in annihilation $\gamma$-rays. Therefore, in this case the analysis similar to that of Ref. [41], would require a specific selection of the source samples.

$n - \bar{n}'$ *mixing.* For a simplicity, we have considered the situation with only $n - n'$ mixing (1), induced via effective D=9 operators (5), which conserves $\tilde{B} = B + B'$. However, there can also exist $n - \bar{n}'$ mixing: there is no fundamental reason to forbid it. However,

the latter mixing, due to the SM structure, emerges from D = 10 operators [85], and depending on the model parameters, $n - \bar{n}'$ mixing mass $\varepsilon_{n\bar{n}'}$ can be much smaller than $n - n'$ mixing mass $\varepsilon$, but can be also comparable to it. In the latter case, with $\varepsilon = \varepsilon_{nn'}$, both the MNS or AMNS could produce the baryon–antibaryon annihilation $\gamma$-rays, without 'help' of the ordinary gas accretion. Interestingly, in the presence of both $\varepsilon_{n\bar{n}'}$ and $\varepsilon_{nn'}$ with the comparable values is not in conflict with the nuclear stability limits, while for the free neutron case it can effectively induce $n - \bar{n}$ oscillation with pretty large rates provided that experimental conditions are properly tuned [85].

$n - n'$ *mass splitting;* in the minimal situation, when $n$ and $n'$ have exactly the same masses, the experimental bounds [18–24] imply $\varepsilon < 10^{-15}$ eV or so. In this case the time of $n - n'$ transition (19) is much larger than the Universe age, and thus it should be an ongoing process in the existing neutron stars (or M neutron stars). However, much larger values of $\varepsilon$ are allowed by the experiment if $n$ and $n'$ are not degenerate in mass. In particular, $n - n'$ oscillation, e.g., with $\varepsilon \sim 10^{-10}$ eV or so can solve the neutron lifetime problem, the $4\sigma$ discrepancy between the neutron lifetimes measured via the bottle and beam experiments, provided that $n$ and $n'$ have a mass splitting $m_{n'} - m_n \sim 1$ $\mu$eV or so [104]. In fact, mass splitting will emerge in models in which mirror parity is spontaneously broken [47] but with a rather small difference between the O and M Higgs VEVs $\langle \phi \rangle$ and $\langle \phi' \rangle$ [105], or it can also emerge dynamically in bigravity scenarios [62,63]. In this case $n - n'$ conversion time will be much smaller, $\tau_\varepsilon < 10^6$ yr or so, so that the most of existing NS should be already transformed in maximally mixed stars with equal amounts and equal radii of the O and M components [31]. Hence, half of the AMNS mass in this case will be constituted by our antinuclear matter, and the $\gamma$-ray emission rate due to accretion will be given by Equation (24).

The alternative to our mechanism can be the existence of antimatter stars (antistars) [106]. The commonly accepted baryogenesis mechanisms fix the value as well as the sign of the BA universally in the whole Universe. In addition, the observations rule out the existence of significant amount of antimatter on scales ranging from the solar system to galaxy and galaxy clusters, and even at very large scales comparable to the present horizon [107,108]. However, more exotic baryogenesis mechanisms (for a review see [109]) can in principle allow the existence of small domains at well-tempered scales in which antimatter could survive in the form of anistars [110–113]. In particular, the Affleck–Dine mechanism [114] can be extended by the coupling of the Affleck–Dine B-charged scalar field to the inflaton [115]. This modification, with properly tuned parameters, can give rise to large baryon overdensities at needed scales in which the stars of specific pattern (or the baryon-dense objects (BDO) as they were named in Ref. [116]) can be formed. In addition, in these overdensities the difference between the baryon and antibaryon amounts can be non-vanishing, and it could be positive as well as negative. Provided that part of the BDO consisting of antibaryons survive in the Milky Way (MW) halo up to present days, they can be observed as the emitters of the $p\bar{p}$ annihilation $\gamma$-rays.

In principle, the AMNS can be distinguished from antistars by the spectral shape of the annihilation $\gamma$-rays. Namely, the proton annihilation on the antistar surfaces should produce $\gamma$-rays with typical spectrum peaked at 70 MeV or so [43]. In the case of the AMNS, the spectral shape will be deformed by the surface redshift effect, by a factor $\exp[\phi]$, where $\phi = \phi(r)$ is the gravitational potential at the surface of antimatter core inside the AMNS. This will rescale down the spectral shape by $(15 - 30)\%$ depending on the AMNS mass, the EoS specifics and the radius of antimatter core $r < R$. In addition, one has to take into account the energy blueshift of the accreted protons: in fact, at the core surface they will be semi-relativistic, with the speeds nearly approaching the speed of light .

In addition, let us recall that the AMNS can radiate substantial energy (20) via the photons in the far UV/soft X-ray ranges which can be an additional tracer for their identification. On the other hand, also antistars can produce X-rays by the mechanism discussed in Ref. [117]. Let us remark, however, in Ref. [41] the sources possibly associated with X ray pulsars were excluded from the possible candidates.

The AMNS could be also observable as ordinary pulsars, if large ordinary magnetic fields are somehow developed in their antibaryon cores. This could be realized, for example, if along with $nn'$ mixing, there is also a kinetic mixing between the O and M photons [65] which effectively renders the mirror electrons and protons mini-charged (with tiny ordinary electric charges). The value of these electric mini-charges are severely restricted by the the cosmological [118] and experimental [119] bounds. Nevertheless, their existence can be effective. Since the antimatter core in the AMNS should consist of the heavy antinuclei and positrons, the AMNS rotation can induce circular electric currents in its antimatter core by the drag mechanism [69] which can be sufficient for these cores to acquire a significant magnetic field, as was discussed in Ref. [31]. Therefore, the AMNS could mimic ordinary pulsars, perhaps with some unusual properties. Having this in mind, maybe the pulsars should not be excluded from the candidate selection provided that their $\gamma$-emission has no high energy tail above 1 GeV or so.

**Funding:** The work was supported in part by the research grant "The Dark Universe: A Synergic Multimessenger Approach" No. 2017X7X85K under the program PRIN 2017 funded by the Ministero dell'Istruzione, Università e della Ricerca (MIUR), and in part by Shota Rustaveli National Science Foundation (SRNSF) of Georgia, grant DI-18-335/New Theoretical Models for Dark Matter Exploration.

**Acknowledgments:** I thank Igor Tkachev for directing my attention to Ref. [41] and Alessandro Drago for useful discussions.

**Conflicts of Interest:** The author declares no conflict of interest.

## Abbreviations

The following abbreviations are used in this manuscript:

| | |
|---|---|
| CSV | Comma-Separated Values |
| EGM | Earth Gravitational Model |
| EGM2008 | Earth Gravitational Models of 2008 |
| GEBCO | The General Bathymetric Chart of the Oceans |
| GIS | Geographic Information System |
| GMT | Generic Mapping Tools |
| QGIS | Quantum GIS |
| IRIS | Incorporated Research Institutions for Seismology |
| NGA | National Geospatial-Intelligence Agency |
| NGDC | National Geophysical Data Center |
| NOAA | National Oceanic and Atmospheric Administration |
| SIO | Scripps Institution of Oceanography |
| SRTM | Shuttle Radar Topography Mission |
| USGS | United States Geological Survey |

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
