# Peer review of "Antistars or Antimatter Cores in Mirror Neutron Stars?"

_universe, doi:10.3390/universe8060313_

Round 1
Reviewer 1 Report
The paper puts forward a beautiful idea that it is feasible in certain scenarios to create a core of our antimatter gravitationally trapped in the mirror star interior. The annihilation of accreting gas on such antimatter cores could explain gamma-ray candidates recently identified in the Fermi LAT catalog as probable antistars.
The paper gives a profound review of mirror matter as a possible dark matter candidate. It should be published after minor editorial corrections.
Historical comments.
p. 4
The author writes: But the real difference is related to CP-violation which was not yet discovered at the time of original works [58, 59].
-- this is true for [58], but not for [59].
The abstract of paper [59] (which was the first to introduce mirror particles) begins with the words:
"In connection with the discovery of CP violation..."
New references on the topics touched upon by the author. 1) http://adsabs.net/abs/2021MPLA...3650215A Alizzi, Abdaljalel; Silagadze, Z. K. Dark photon portal into mirror world Modern Physics Letters A, Volume 36, Issue 30, id. 2150215 2) https://ui.adsabs.harvard.edu/abs/2022JCAP...03..009B/abstract X-ray signature of antistars in the Galaxy Bondar, A. E.; Blinnikov, S. I.; Bykov, A. M.; Dolgov, A. D.; Postnov, K. A. Journal of Cosmology and Astroparticle Physics, Volume 2022, Issue 03, id.009, 14 pp. 3) http://adsabs.net/abs/2022arXiv220317228Z Zöllner, Rico; Kämpfer, Burkhard Exotic Cores with and without Dark-Matter Admixtures in Compact Stars The author may wish to add a citation or a brief discussion on those papers.Obvious typos.
Abstract.
recently identified in the Fermi LAT catalog, --> change comma to dot.
p. 2
Section III we discusses --> either Section III discusses, or In Section III we discuss
p. 3
Along with the ordinary stars also dark mirror stars can be formed the Galaxy. --> Along with the ordinary stars also dark mirror stars can be formed in the Galaxy.
the first mirror stars start must form --> the first mirror stars must start forming
bullet cluster --> Bullet cluster
Obscure places
p. 3
The helium dominated M stars, being (???) should have...
...coining as M fermions the species with the RH
weak interactions and them (???) positive quantum numbers.
Author Response
I thank to reviewer for carefully reading the manuscript, for correcting my misunderstanding about the time ordering of CP violation discovery and the second paper on mirror particles, indicating some attainable refs. and finding several typos in my text.
I fixed all indicated typos, changed the sentence as "the CP-violation was just discovered at the time of KOP paper", and added indicated refs. in relevant places. I also added some additional refs. on dark matter in the neutron stars which came out after the first version of my paper.
Reviewer 2 Report
The author, in the present work review the possibility of creation a core of antimatter gravitationallytrapped in the mirror star interior. Moreover, the author provides, in this way some explanations about the the origin of $\gamma$-source candidates with unusual spectrum compatible to baryon-antibaryonannihilation and also the production of the flux of cosmic antihelium and also heavier antinuclei which are hunted in the AMS-02 experiment.
The paper is well written and contains several results that will be useful to the scientific community. The presentation is very analytical, covering all the relevant cases.
Where necessary the appropriate references were given. In general, the bibliographyis complete. I consider that the paper deserves publication. However, I consider that it would be useful if the author could add some brief comments to the following issue:According to my opinion the disadvantage of the present work is the lack of presentation of results concerning basic properties (mass, radius, etc.)
of the compact stars under consideration. In other words how (quantitatively and qualitatively) the existence of
an antimmater in their interior can affect their bulk properties properties.
Even more, it needs to be clarified whether the present gravitational wave detection experiments, which are now one of the most powerful tools for detecting the internal structure of compact objects, could shed light on the author's claims.
Author Response
I thank the reviewer for useful comments. In fact, modifications of mass-radius relations for mixed stars were discussed in our paper Berezhiani, Biondi, Mannarelli, Tonelli and I did not deviate here for discussing these issues in depth since it would lead to enlargement of the paper. Moreover, for very slow n-n' transitions ($\tau > 10^{15}$ yr) which I discussed mainly in this paper the presence of few planetary mass dark matter cores would not change much the bulk properties, apart of providing an additional internal heating of the star which is discussed in the above paper and the main arguments are shortly reproduced here. Also gravitational wave signals were discussed in the above paper and I limited myself to refer it for this discussion. Once again, for so slow transition the small amount of dark matter in coalescing neutron stars will change nothing in their deformability and will have practically no impact. In other words, any equation of states gives the NS structure/radii compatible with the 2017 GW signal from the NS merger will remain compatible in my scenario with small dark cores.
Reviewer 3 Report
The author of the article studies the possibility of the existence in the universe of anti-stars or antimatter cores in neutron stars containing mirror matter, represented by hidden parallel sector particles that are exact replicas of ordinary particles.
The author discusses these effects by assuming that the n-n' mixing is due to mass mixing alone. Under these circumstances they obtain that the process of transition to mixed stars is very slow and that therefore the n-n' transition should be a process that is still ongoing in neutron stars or mirror neutron stars.
Nevertheless, the study is interesting as a basis for future extensions of the theory.
Author Response
I thank referee for this review. He correctly notes that I have mainly discussed slow n-n' transitions which transitions are still ongoing in existing neutron stars. In this case the limits on the NS surface temperatures leads to a limit on the transition time $\tau > 10^{15}$ yr or so, at least four orders of magnitude larger than the age of the Universe.
But the transitions can be also rather fast, say with $\tau < 10^{5}$ yr or so in which case for any NS older than million years the transition is over, the maximally mixed configuration is reached and no more internal heat is produced by n-n' conversion. The bulk properties (M/R relations etc.) of maximally mixed stars were discussed in my previous paper (Berezhiani, Biondi et al.) and I just shortly discussed this possibility in section VI of the present manuscript, as well as possibility of fast transition induced by non-diagonal magnetic moment between n and n' states.
Reviewer 4 Report
The manuscript presents a detailed review of mirror dark matter and its possible interaction channels with visible matter. Special attention is paid to neutron stars in which the transition between neutron and mirror neutron could occur causing the creation of antimatter core inside the star. I consider the paper to be very well written and very pedagogical. It could be useful both for readers specialized in the equation of state and for readers with more astrophysical background. In my opinion, the paper merits publication and I recommend it for publication in Universe.
Below there are some typos:
1) page 6, 1 column, above Eq. 7, should be N=\kappa (M/m)
2) page 6, 2 column, above Eq. 9, 'then N'(t) increases while N(t)...'
3) page 6, 2 column, above Eq. 10, 'composition which itself evolves...'
4) page 7, 1 column, above Eq. 14, 'Hence, a small...
Author Response
I thank the referee for a nice review, and for indicating several typos.
I fixed these typos and also some few other ones.